

# Fracton-elasticity duality of two-dimensional superfluid vortex crystals: defect interactions and quantum melting

**Dung Xuan Nguyen** [1*]**, Andrey Gromov**[1] **and Sergej Moroz**[2,3]

**1** Brown Theoretical Physics Center and Department of Physics,
Brown University, 182 Hope Street, Providence, RI 02912, USA
**2** Physik-Department, Technische Universität München, D-85748 Garching, Germany
**3** Munich Center for Quantum Science and Technology (MCQST),
Schellingstr. 4, D-80799 München, Germany

* dungmuop@gmail.com

## Abstract

Employing the fracton-elastic duality, we develop a low-energy effective theory of a zero-temperature vortex crystal in a two-dimensional bosonic superfluid which naturally incorporates crystalline topological defects. We extract static interactions between these defects and investigate several continuous quantum transitions triggered by the Higgs condensation of vortex vacancies/interstitials and dislocations. We propose that the quantum melting of the vortex crystal towards the hexatic or smectic phase may occur via a pair of continuous transitions separated by an intermediate vortex supersolid phase.

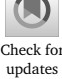

## Contents

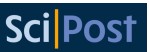
## 1  Introduction

Vortices in superfluids are characterized by the quantized circulation of superfluid velocity and thus manifest quantum mechanics on macroscopic scales. The physics of vortices proved to be central for the understanding of numerous phenomena in superfluids such as turbulence [1,2], dissipation [3,4], the thermal Berezinskii-Kosterlitz-Thouless (BKT) and the quantum Mott-superfluid phase transitions [5–10]. Today vortex matter in superfluids is a broad and active area of experimental and theoretical research [11,12].

As first predicted by Abrikosov [13], quantum vortices form regular crystals in type-II superconductors in an external magnetic field. In neutral superfluids similar vortex crystals emerge under an external rotation [14–17]. These were first investigated theoretically by Tkachenko, who treated the problem in the incompressible limit [18–20]. He discovered that the triangular lattice is favored energetically and determined the nature of collective excitations which are known today as Tkachenko waves [21]. These predictions were later supported by the hydrodynamic approach [22,23]. Compressible rotating superfluids exhibit Tkachenko waves with a soft quadratic dispersion at low momenta. These were investigated theoretically [24–26] and experimentally [27]. The softness of the Tkachenko mode implies that true off-diagonal long-range $U(1)$ order is destroyed by quantum fluctuations in two-dimensional vortex crystals at vanishing temperature and the system exhibits only algebraically decaying $U(1)$ order [28–30]. More recently, low-energy effective field theories of vortex crystals of two-dimensional superfluids were developed [31–33]. These shed new light on the nature of spontaneous symmetry breaking and Hall responses in this system.

While the physics of the two-dimensional superfluid vortex crystal phase is rather well-understood, outstanding unsolved questions in this field are concentrated around thermal and quantum meltings of the vortex lattice. Although the Abrikosov mean-field theory predicts a direct second-order transition between the vortex crystal in a superconductor and the normal phase [13], fluctuations are expected to invalidate this result [34]. It was proposed already in [35] that, at a sufficiently large temperature, a vortex crystal in a two-dimensional superconductor should melt into a vortex fluid via a pair of BKT-like phase transitions triggered by unbinding of dislocation and disclination defects of the crystal. New experiments with clean superconducting films appear to agree with this scenario [36,37]. On the other hand, extensive theoretical investigations of this problem [38–44] gave contradictory predictions for the nature of the thermal melting phase transition of vortex crystals. Most studies indicated a weakly first-order thermal transition towards an isotropic vortex fluid, see [45] for a review. When the density of quantum vortices becomes of the order of the density of elementary boson particles, the vortex crystal at zero temperature is believed to undergo a quantum melting transition into a strongly correlated vortex fluid phase. Various estimates based on the Lindemann criterion, reviewed in [17], predict for bosons with short-range interactions the transition to happen at

filling[1] $\nu \sim 10$. Early exact diagonalization calculations [46] are consistent with this result, while the later study [47] indicates that the vortex lattice might survive even at the filling factor $\nu = 2$, where it energetically competes with the quantum smectic phase. Despite all these results, it is fair to say that in the thermodynamic limit, the nature of the quantum melting transition and the resulting quantum vortex fluid phase(s) (at fillings close to the critical) are currently not well understood.

It is well known that proliferation of topological defects induces thermal melting of a two-dimensional crystal. The idea was first suggested by Kosterlitz and Thouless in [9, 10], where they used renormalization group (RG) to analyze the dislocation mediated melting. An important next step was done by Halperin and Nelson [48] who extended this approach by incorporating also disclinations. Later on, Young [49] performed the RG calculation to derive the correlation length critical exponent near the crystalline-hexatic melting transition.

Since it is expected that dislocations and disclinations play an important role in thermal and quantum melting of two-dimensional vortex crystals, it is desirable to develop a theory, where these topological defects are naturally incorporated. In time-reversal invariant crystals, such a formalism was first developed by Kleinert [50–53], who rewrote the theory of classical elasticity as a dual gauge theory with symmetric tensor gauge fields. Under this duality, topological defects are mapped to matter fields charged under the dual gauge fields. This duality was extended to the quantum realm in [54], and the general theory was developed in [55, 56].

In an unrelated line of work, a new type of topological phases of matter [57–60] was discovered. These phases are characterized by the presence of local excitations, referred to as *fractons*, that cannot freely move through space – the property that is often referred to as restricted mobility. In a parallel yet unrelated line of work, a new class of algebraic spin liquids has been discovered [61–63]. At long distances, these spin liquids were described by an emergent Abelian gauge theory reminiscent of linearized Horava-Lifshitz gravity [62]. It was noticed by Pretko [64, 65] that gauge theories of [61–63] *must* couple to particles with restricted mobility. The restricted mobility effect follows directly from the Gauss law constraint of these theories. These gauge theories, when considered in two spatial dimensions, are precisely dual to the two-dimensional elasticity [66–69]. Indeed, crystalline defects in the quantum theory of elasticity are characterized by restricted mobility: dislocations can only move along their Burgers vector, while disclinations are immobile. Moreover, a disclination dipole is equivalent to a dislocation, with the Burgers vector perpendicular to the dipole moment. The duality has been generalized in several different directions and was used to revisit the melting transition in quantum crystals [70–73]. It was also generalized to three spatial dimensions [74] and to quantum smectic phases [75], that are dual to more general multipole theories studied in [76–78].

In addition to the proliferation mechanism of dislocations and disclinations, the physics of vacancies and interstitials plays a central role in the quantum melting of solids [68, 70, 71]. Specifically, if these defects carry a finite charge under a global symmetry, dislocations cannot proliferate unless that symmetry is broken spontaneously, which requires proliferation of vacancy/interstitial defects. In the context of finite-temperature three-dimensional vortex crystals in charged superconductors entropic proliferation of vacancies and interstitials has been discussed in detail already in [79, 80]. More recently, Ref. [71] briefly discussed condensation of these defects in zero-temperature two-dimensional superfluid vortex crystals.

In the present work, we utilize the duality between crystalline defects and fractons to study the vortex crystals and their quantum melting transitions. Our main findings are:

- We extract static interactions between topological defects of the vortex crystal phase such as vortex defects (vacancies/interstitials) and dislocations. We discover that the

---

[1] In a homogeneous bosonic system the filling factor $\nu$ is defined as the ratio of the density of bosons $n_b$ to the density of vortices $n_v$.

vortex-vortex interaction is generically short-ranged due to screening caused by the vortex crystal, see Eq. (4.3). Similar to the inter-vortex potential in superconductors, this interaction can be either repulsive (type-II) or attractive (type-I). The sign of the interaction depends on a linear combination of the two elastic moduli of the vortex crystal. The inter-dislocation interaction that we find agrees with the previous work [81]. Finally, we compute the interaction between a vortex vacancy/interstitial and a dislocation (4.7) and find that at distances much larger than the screening length, it is identical to the electrostatic dipole-charge interaction in two spatial dimensions.

- Since the vortex number is conserved in the superfluid phase, in the presence of interstitials/vacancies, the vortex crystal must satisfy the modified glide constraint (3.17). In addition, a bound state of two dislocations of opposite charge carries the vortex number charge and cannot unbind without condensing vortices. These two observations imply that a conventional continuous dislocation-assisted quantum melting transition between the vortex crystal and a hexatic or smectic phase is not allowed [71]. As a result, the quantum melting transition is either a first-order phase transition or two conventional continuous quantum transitions separated by an intermediate phase.[2] In this paper, we restrict our attention to the latter two-step scenario, see Fig. 1. First, we study the precursor of the quantum melting and investigate a direct continuous quantum transition between the vortex crystal and a chiral crystal that we find to satisfy the ordinary glide constraint. The transition is driven by condensation of vortex vacancies/interstitials, which is governed by the abelian Higgs mechanism. Since this phase exhibits translation symmetry breaking simultaneously with vortex defect condensation, we will refer to it as the vortex supersolid crystal[3]. We determine the relationship between the elastic moduli of the vortex supersolid crystal and the vortex crystal. Second, we analyze the continuous quantum melting of the vortex supersolid crystal governed by the proliferation of dislocations. We investigate two different possible Higgs transitions and derive the effective gauge theories of resulting quantum nematic and smectic phases.

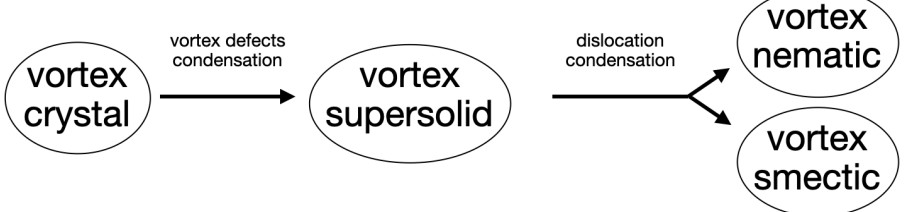

Figure 1: A two-step continuous quantum melting scenario of the two-dimensional vortex crystal to the vortex nematic or smectic phase via the intermediate vortex supersolid phase.

## 2 Effective theory of vortex crystal

The starting point of our investigation is the effective field theory of a two-dimensional vortex crystal derived in [32, 33]. Its low-energy degrees of freedom are the dynamical $u(1)$

---

[2]One can also envision an unconventional continuous melting transition where both vortex defects and dislocations are proliferated simultaneously, but we will not investigate such possibility in this paper.

[3]In charged three-dimensional superconductors a related finite-temperature phase was investigated in detail in [79, 80]. In that problem, it is related via the boson-vortex duality to the quantum supersolid phase of short-range interacting bosons [82].

gauge field $a_\mu$ (which via the boson-vortex duality encodes the physics of the coarse-grained superfluid degree of freedom) and coarse-grained displacements of vortices $u^i$ from their equilibrium crystal positions. It will be sufficient to work with the quadratic Lagrangian which has the form

$$\mathcal{L} = \mathcal{L}_g(a_\mu) - \frac{B_0 n_0}{2}\epsilon_{ij}u^i\partial_t u^j - \frac{1}{2}C_{ij;kl}u_{ij}u_{kl} + B_0 e_i u^i + a_\mu j_v^\mu. \tag{2.1}$$

Specifically, in this paper $a_\mu$ denotes the $u(1)$ gauge field fluctuation around a static background $\bar{a}_\mu$ which produces a magnetic field fixed by the superfluid density in the ground state, i.e., $n_0 = \epsilon_{ij}\partial_i\bar{a}_j$. The quadratic Lagrangian $\mathcal{L}_g(a_\mu)$ governs the dynamics of the gauge field fluctuation $a_\mu$. For now its concrete form is unimportant for us, but see [32,33] for the specific examples. The second term fixes the dynamics of quantum vortices and ensures that in a superfluid of a fixed density $n_0$ subject to a constant magnetic field[4] $B_0$ the cartesian components of the displacement $u^x$ and $u^y$ form a canonically conjugate pair of variables. The next term depends on the strain $u_{ij} = (\partial_i u_j + \partial_j u_i)/2$ and fixes the potential elastic energy of the two-dimensional vortex crystal, which henceforth is assumed to be triangular, implying the elasticity tensor of the form[5]

$$C_{ij;kl} = 8C_1 P^{(0)}_{ij;kl} + 4C_2 P^{(2)}_{ij;kl}, \tag{2.2}$$

where we introduced the compression and shear projection operators [56]

$$P^{(0)}_{ij;kl} = \frac{1}{2}\delta_{ij}\delta_{kl}, \tag{2.3}$$

$$P^{(2)}_{ij;kl} = \frac{1}{2}\left(\delta_{ik}\delta_{jl} + \delta_{il}\delta_{jk} - \delta_{ij}\delta_{kl}\right). \tag{2.4}$$

In contrast to an ordinary crystal, in the vortex crystal the compression modulus $C_1$ does not have to be positive [18–20,26,81]. On the other hand, the shear modulus $C_2 > 0$. The derivation of the constraints on the elastic moduli is presented in Appendix A, where we perform the stability analysis of the quadratic effective theory. In the Lagrangian (2.1) vortices couple to the superfluid only via the dipole term $\sim e_i u^i$, where the electric field[6] $e_i = \partial_t a_i - \partial_i a_t$. Finally, the last term in the Lagrangian allows us to incorporate the vortex defects, such as vacancies and interstitials, on top of the vortex lattice, whose current $j_v^\mu$ couples minimally to the $u(1)$ gauge field. Importantly, as will be explained in some detail in Sec. 5, in this paper vortex defects couple only to the fluctuation $a_\mu$, but not to the background $\bar{a}_\mu$.

Notice that $C_{ij;kl}$ is symmetric under both $i \leftrightarrow j$ and $k \leftrightarrow l$ exchanges. Consequently, the stress tensor

$$\sigma_{ij} = -\frac{\delta\mathcal{S}}{\delta(\partial_i u_j)} \tag{2.5}$$

is symmetric in space indices

$$\epsilon_{ij}\sigma_{ij} = 0, \tag{2.6}$$

which is sometimes referred to as the Ehrenfest constraint [56]. We can rewrite the symmetric strain as $u_{ij} = \partial_i u_j - \theta\epsilon_{ij}$ where $\theta$ is the bond angle

$$\theta = \frac{1}{2}\epsilon_{ij}\partial_i u_j. \tag{2.7}$$

In the next section, we will treat $\theta$ as an independent field. It enters the action as a Lagrange multiplier which enforces the Ehrenfest constraint.

---

[4]In a neutral superfluid an effective constant magnetic field can be realized by external rotation or with artificial gauge fields, see [15–17,83].

[5]Our notation for elastic moduli $C_1$ and $C_2$ follows Refs. [26,32,33,81].

[6]From the boson-vortex duality the superfluid current $j^\mu = \epsilon^{\mu\nu\rho}\partial_\nu a_\rho$. This implies that the electric field $e_i$ is equal in magnitude and is perpendicular to the superfluid current $j^i$.

# 3 Symmetric tensor gauge theory of vortex crystal elasticity

In this section we will apply elastic duality transformation to the vortex lattice theory (2.1). For ordinary crystals, the formalism was introduced first by Kleinert [50–53] to study the thermal melting and then adapted to supersolids [54] and to quantum liquid crystals [56]. This formalism was revised recently [66–68, 71] in the context of duality between crystalline defects and fractons.

First, we introduce the Hubbard-Stratonovich (HS) fields $\tilde{\pi}_i$ and $\tilde{\sigma}_{ij}$ and obtain the equivalent Lagrangian

$$\mathcal{L} = \mathcal{L}_g(a_\mu) + \frac{1}{2B_0 n_0}\epsilon^{ij}\tilde{\pi}_i\partial_t\tilde{\pi}_j + \tilde{\pi}_i\partial_t u_i + \frac{1}{2}C_{ij;kl}^{-1}\tilde{\sigma}_{ij}\tilde{\sigma}_{kl} - \tilde{\sigma}_{ij}(\partial_i u_j - \theta\epsilon_{ij}) + B_0 e^i u_i + a_\mu j_\nu^\mu, \quad (3.1)$$

where we denoted [56]

$$C_{ij;kl}^{-1} = \frac{1}{8C_1}P_{ij;kl}^{(0)} + \frac{1}{4C_2}P_{ij;kl}^{(2)}, \quad (3.2)$$

as the formal inversion of $C_{ij;kl}$, which is defined as $C_{ij;kl}^{-1}C_{kl;mn} = P_{ij;mn}^{(0)} + P_{ij;mn}^{(2)}$. Integrating out the HS fields in (3.1) gives us back the Lagrangian (2.1). Next we separate the fields $u_i$ and $\theta$ into singular and (smooth) elastic parts as follows

$$u_i = u_i^s + u_i^e, \qquad \theta = \theta^s + \theta^e. \quad (3.3)$$

The singular part of the strain $u_{ij}^s = \partial_i u_j^s - \epsilon_{ij}\theta^s$ is related to the density of crystalline defects. Integrating out the elastic components gives us the conservation law

$$-\partial_t\tilde{\pi}_i + \partial_j\tilde{\sigma}_{ji} + B_0(\partial_t a_i - \partial_i a_t) = 0, \quad (3.4)$$

and the Ehrenfest constraint

$$\epsilon^{ij}\tilde{\sigma}_{ij} = 0. \quad (3.5)$$

We now define

$$\pi_i = \tilde{\pi}_i - B_0 a_i, \qquad \sigma_{ij} = \tilde{\sigma}_{ij} - B_0\delta_{ij}a_t, \quad (3.6)$$

and introduce *elastic-electromagnetic* fields, which are determined by $\pi_i$ and $\sigma_{ij}$

$$B^i = \epsilon^{ij}\pi_j, \qquad E^{ij} = \epsilon^{ik}\epsilon^{jl}\sigma_{kl} \quad (3.7)$$

to rewrite (3.4) and (3.5) as

$$\partial_t B^i + \epsilon_{jk}\partial^j E^{ki} = 0, \quad (3.8)$$

$$\epsilon^{ij}E_{ij} = 0. \quad (3.9)$$

The Bianchi identity (3.8) and the Ehrenfest constraint (3.9) are satisfied automatically if $B^i$ and $E_{ij}$ are expressed in terms of a symmetric tensor gauge field $A_{ij}$ and a scalar potential $\varphi$

$$B^i = \epsilon_{jk}\partial^j A^{ki}, \quad (3.10)$$

$$E_{ij} = -\partial_t A_{ij} - \partial_i\partial_j\varphi. \quad (3.11)$$

After all the above manipulations, we obtain the dual Lagrangian

$$\mathcal{L} = \mathcal{L}_g(a_\mu) + \frac{1}{2B_0 n_0}\epsilon_{ij}(B^i + B_0\epsilon^{ik}a_k)\partial_t(B^j + B_0\epsilon^{jl}a_l)$$

$$+ \frac{1}{2}\tilde{C}_{ij;kl}^{-1}(E^{ij} + B_0\delta^{ij}a_t)(E^{kl} + B_0\delta^{kl}a_t) + A_{ij}J^{ij} + \varphi\rho + a_\mu j_\nu^\mu, \quad (3.12)$$

with[7] $\tilde{C}^{-1}_{ij;kl} = \epsilon^{ii'}\epsilon^{ji'}\epsilon^{ki'}\epsilon^{li'}C^{-1}_{i'j';k'l'}$. We also defined the dislocation current

$$J^{ij} = \epsilon^{il}\epsilon^{jk}(\partial_l\partial_t - \partial_t\partial_l)u^s_k, \tag{3.13}$$

and the disclination density

$$\rho = \rho_\theta + \epsilon^{ij}\partial_i\chi_j, \tag{3.14}$$

where we have introduced the Burgers vector density of dislocation $\chi_i = \epsilon^{lj}\partial_l\partial_j u^s_i$. Above the disclination density was separated into the "free" part $\rho_\theta = \epsilon^{ij}\partial_i\partial_j\theta^s$ and the part coming from gradients of the dislocation density $\epsilon^{ij}\partial_i\chi_j$ [84].

We point out that the coupling between the tensor elastic gauge fields and the $u(1)$ gauge field in Eq. (3.12) is very different to the axion-like coupling found recently in the dual elasticity theory of a time-reversal invariant supersolid [68, 70, 71]. We attribute this to a fundamental difference in the symmetry breaking pattern of the vortex crystal and the supersolid.

The dual theory (3.12) is invariant under the following gauge transformations

$$A_{ij} \to A_{ij} + \partial_i\partial_j\alpha + B_0\delta_{ij}\beta, \qquad \varphi \to \varphi - \partial_t\alpha, \qquad a_\mu \to a_\mu + \partial_\mu\beta. \tag{3.15}$$

In addition to the conservation of the disclination density

$$\partial_t\rho + \partial_i\partial_j J^{ij} = 0, \tag{3.16}$$

the gauge symmetry implies the modified glide constraint

$$B_0\delta_{ij}J^{ij} - \partial_\mu j^\mu_v = 0. \tag{3.17}$$

This deserves a clarification: In absence of interstitials and vacancies, in classic elasticity dislocations satisfy an ordinary glide constraint [85] which is encoded in the relation

$$\delta_{ij}J^{ij} = 0. \tag{3.18}$$

Here in addition to vortices forming the periodic crystal, we also include the vortex interstitial and vacancy excitations on top of the crystal with the density $j^0_v$. In the vortex crystal the total vortex number is conserved which ensures that to the glide constraint (3.18) is modified to (3.17).

Finally, we compute excitation modes of the vortex crystal encoded in the dual Lagrangian (3.12). To this end one must specify the superfluid part of the Lagrangian $\mathcal{L}_g(a_\mu)$. Following [32], we first consider only the non-dynamical magnetic Lagrangian $\mathcal{L}_g(a_\mu) = -mc_s^2 b^2/2n_0$, where $m$ denotes the mass of the elementary boson, $c_s$ is the velocity of sound and $b = \epsilon^{ij}\partial_i a_j$. This term is of leading-order in the derivative expansion developed in [32] and thus contains information about the low-energy and low-momentum part of the energy spectrum. By solving equations of motion derived from the Lagrangian (3.12) we find the Tkachenko mode with the quadratic dispersion relation [26]

$$\omega^2 = \frac{2mC_2 c_s^2}{B_0^2 n_0}k^4. \tag{3.19}$$

It is straightforward now to add the electric term $me^2/(2n_0)$ to the Lagrangian $\mathcal{L}_g(a_\mu)$ which within the derivative expansion constitute a next-to-leading order correction. In addition to higher-momentum corrections to the Tkachenko dispersion, this term gives rise to the gapped Kohn mode in the energy spectrum, which is in agreement with Ref. [32]. As discussed in Ref. [33], in addition to the electric term $me^2/(2n_0)$ the $u(1)$ gauge theory contains another

---

[7]Notice that with the definition (3.2), one can check that $\tilde{C}^{-1}_{ij;kl} = C^{-1}_{ij;kl}$.

next-to-leading contribution: a time-reversal breaking Berry term which survives in the lowest Landau level limit ($m \to 0$) and is entirely responsible for the gauge field dynamics in that regime.

We notice that the dual theory (3.12) exhibits a global shift symmetry $B^i \to B^i + c^i$ which corresponds to the magnetic translation symmetry of a time-reversal breaking crystal in the dual description. This symmetry prohibits the quadratic term $B^i \delta_{ij} B^j$ to appear in the dual quadratic Lagrangian and thus protects the quadratic dispersion relation of the soft Tkachenko mode from linear corrections at low momenta.

## 4 Static interactions between topological defects in vortex crystal phase

In this section we will extract static interaction potentials between different defects of the vortex crystal from the dual theory (3.12). The relevant static part of the dual Lagrangian is

$$\mathcal{L}_{st} = \frac{m(\partial_i a_t)^2}{2n_0} + \frac{1}{2}(\tilde{C}^{-1})^{ijkl}(\partial_i \partial_j \varphi + B_0 \delta_{ij} a_t)(\partial_k \partial_l \varphi + B_0 \delta_{kl} a_t) + \varphi \rho + a_t \rho_v. \quad (4.1)$$

Here the first term originates from the electric part $me^2/(2n_0)$ of the Lagrangian $\mathcal{L}_g(a_\mu)$. After integrating out the gauge fields $a_t$ and $\varphi$, we find in Fourier space

$$\mathcal{L} = -\frac{1}{2} \begin{pmatrix} \rho(-q) & \rho_v(-q) \end{pmatrix} \begin{pmatrix} \frac{8C_2(B_0^2 n_0 + 4C_1 mq^2)}{q^4(B_0^2 n_0 + 2(2C_1 + C_2)mq^2)} & -\frac{4B_0 C_2 n_0}{q^2(B_0^2 n_0 + 2(2C_1 + C_2)mq^2)} \\ -\frac{4B_0 C_2 n_0}{q^2(B_0^2 n_0 + 2(2C_1 + C_2)mq^2)} & \frac{2(2C_1 + C_2)n_0}{B_0^2 n_0 + 2(2C_1 + C_2)mq^2} \end{pmatrix} \begin{pmatrix} \rho(q) \\ \rho_v(q) \end{pmatrix}, \quad (4.2)$$

which contains all information about static potentials between topological defects. In the crystalline phase pairs of disclinations are tightly confined into dislocations and thus here we consider only vortex vacancies/interstitials and crystal dislocations.

The static potential between two vacancy/interstitial vortex defects is

$$V_v(q) = \frac{2(2C_1 + C_2)n_0}{B_0^2 n_0 + 2(2C_1 + C_2)mq^2} = \frac{n_0/m}{q^2 + \lambda_v^{-2}}, \quad (4.3)$$

where we introduced the vortex screening length $\lambda_v = \sqrt{2(2C_1 + C_2)m/B_0^2 n_0}$.[8] Since in the vortex crystal the combination $2C_1 + C_2$ can be either positive or negative [26, 81], see Section A, we should distinguish three different cases. For $2C_1 + C_2 > 0$ the inter-vortex potential is repulsive and falls off as $K_0(r/\lambda_v)$. It is screened due to the presence of the elastic vortex crystal. As a result it does not decay logarithmically, as in a non-rotating superfluid, but exponentially at distances much larger than $\lambda_v$. Formally, it is identical to interaction of vortices in a type-II superconductor. For $2C_1 + C_2 = 0$ we have $\lambda_v = 0$ and the inter-vortex potential disappears. This is similar to what happens if a superconductor is tuned to the Bogomolny point [12].[9] For $2C_1 + C_2 < 0$ the screening length becomes imaginary and we cannot trust our quadratic theory at $q \sim \lambda_v^{-1}$. Within the regime of validity, i.e., for $q \ll \lambda_v^{-1}$, the intervortex potential is now attractive similar to what happens in a type-I superconductor.

We can extract the dislocation-dislocation static potential by assuming there are no free disclinations, i.e., $\rho_\theta = 0$, and thus $\rho = \epsilon^{ij} \partial_i \chi_j$. Given that, in momentum space we obtain

---

[8]A similar length scale was introduced in [81]. According to that paper $\lambda_v^2$ changes sign as one crosses over from the incompressible Tkachenko regime to the compressible lowest Landau level regime of the vortex crystal.

[9] Note that in this regime the actual interaction between vortices is fixed by higher derivative terms that were not incorporated into the dual Lagrangian (3.12).

the inter-dislocation Lagrangian

$$\mathcal{L}_{disl} = -\frac{1}{2}\chi_T^i(-q)\left[\frac{8C_2}{q^2} - \frac{C_2}{2C_1 + C_2}\frac{8C_2}{(q^2 + \lambda_v^{-2})}\right]\chi_T^i(q), \tag{4.4}$$

where $\chi_T^i(q) = (\delta^{ij} - q^i q^j/q^2)\chi_j(q)$. This result is in agreement with the previous work [81].[10] In the limit $\lambda_v^{-1} = 0$, Eq. (4.4) reduces to the well-known static interaction between two dislocations in a time-reversal invariant two-dimensional crystal [84]. At momenta much smaller than the inverse of the vortex screening length $\lambda_v$ our result simplifies to

$$\mathcal{L}_{disl} = -\frac{1}{2}\chi_T^i(-q)\frac{8C_2}{q^2}\chi_T^i(q). \tag{4.5}$$

Finally, we find a long-range static vacancy/interstitial-dislocation interaction

$$\mathcal{L}_{v-disl} = -\frac{1}{2}\rho_v(-q)V_i(q)\chi_i(q), \tag{4.6}$$

with

$$V_i(q) = i\epsilon_{ij}q_j\frac{4\lambda_v^{-2}C_2/B_0}{q^2(q^2 + \lambda_v^{-2})}. \tag{4.7}$$

At distances much larger than the vortex screening length this is identical with the electrostatic dipole-charge potential in two spatial dimensions.

# 5   Higgs condensation of vortex vacancy/interstitial defects

A conventional continuous dislocation-mediated quantum transition between a superfluid vortex crystal and a vortex hexatic or smectic phase is not possible. Due to the conservation of the vortex number, one cannot proliferate dislocations and restore translation symmetry without simultaneously proliferating quantum vortex defects [71]. This is reflected in the form of the modified glide constraint (3.17). Mathematically, the problem is that in the vortex crystal phase, dislocations couple non-locally to the long-range gauge field $a_\mu$ and so one cannot condense them. In this section, we investigate the Higgs condensation of vortex vacancy/interstitial defects and find that the vortex crystal transforms into a vortex supersolid chiral crystal, which obeys the ordinary glide constraint. Condensation of vortex defects in the vortex crystal has been recently studied in Ref. [71]. Our findings, however, somewhat differ from [71], and we discuss the differences at the end of this section.

In an ordinary bosonic crystal an isolated vacancy defect costs a positive elastic energy $E_v$, but since it is mobile and can hop on a lattice, its energy spectrum forms a band whose minimum has an energy that is lower than $E_v$. If the energy of the band minimum is lower than zero, the crystal ground state becomes unstable and condensation of the bosonic vacancy defects occurs resulting in a supersolid ground state [87]. Similar arguments can be applied to interstitial defects. In this section we extend the mechanism presented above to two-dimensional vortex crystals in bosonic superfluids.

To condense vortex vacancy/interstitial defects, first, we must replace $a_\mu j_v^\mu$ in the Lagrangian (3.12) by a low-energy effective Lagrangian that depends on the vortex defect bosonic

---

[10]To make a direct comparison with Eq. (18) in [81] one must Fourier transform (4.4) to position space using for example Appendix A of [86].

field $\psi_v$. Microscopically, the vortex Lagrangian can be fixed by studying vacancies and interstitials hopping in the vortex crystal. Specifically, one can adopt the tight-binding approximation for the hopping defects, determine their band structure and expand the resulting dispersion around its minimum. Vacancies and interstitials must be treated separately since the former hop on the original triangular lattice, whereas the latter live on the dual (honeycomb) lattice. In this paper we will restrict our discussion to condensation of vacancies, but very similar arguments apply if vortex interstitials condense instead. Since vortices see superfluid density as a magnetic field and assuming a constant dual background field $\bar{b} = \epsilon_{ij}\partial_i \bar{a}_j = n_0$ in the ground state, the band structure of the vortex vacancy quasiparticles must be determined from the solution of the corresponding Hofstadter problem on a triangular lattice [88]. This is extremely sensitive to the dual magnetic flux per plaquette. To illustrate the main idea, here we will work at an even-integer-valued filling $v = n_0/n_v$, where $n_v = B_0/2\pi$ is the vortex density in the vortex crystal ground state. This corresponds to the integer number of dual magnetic flux quanta piercing a unit triangular plaquette[11], which is *unobservable* by the hopping vortex vacancy.[12] The resulting energy band has a minimum at the zero-momentum $\Gamma$ point, and we will expand its dispersion relation to quadratic order in momentum around it. Assuming some additional short-range repulsive interactions between the vacancy quasiparticles, we arrive at the following Lagrangian encoding the physics of vacancies

$$\mathcal{L}_v = i\psi_v^\dagger D_t \psi_v - \frac{1}{2m_v}|D_i\psi_v|^2 - \mu_v \psi_v^\dagger \psi_v - \frac{g_v}{2}(\psi_v^\dagger \psi_v)^2, \qquad D_\mu = \partial_\mu - iq_v a_\mu, \qquad (5.1)$$

where $q_v = -2\pi$ is the charge of the elementary vacancy with respect to the $u(1)$ gauge field. Importantly, the vortex vacancy couples minimally *only* to the fluctuation $a_\mu$ of the gauge field, but not to the background $\bar{a}_\mu$.

We are interested here in the Higgs transition induced by the vortex vacancy condensation as one changes the vacancy chemical potential $\mu_v$ from positive to negative values. Writing $\psi_v = \sqrt{\bar{\rho}_v}e^{i\varphi_v}$ and following Appendix B, the Lagrangian of condensed vacancies reduces to the quadratic form

$$\mathcal{L}_v = q_v a_t \bar{\rho}_v + \frac{1}{2g_v}(\partial_t \varphi_v - q_v a_t)^2 - \frac{\bar{\rho}_v}{2m_v}(\partial_i \varphi_v - q_v a_i)^2, \qquad (5.2)$$

where the phase field $\varphi_v$ is regular.

Now we consider the symmetric tensor gauge theory (3.12) derived in Sec. 3 and replace $a_\mu j_v^\mu$ with the Lagrangian (5.2). Performing the regular gauge transformation $\tilde{a}_\mu = a_\mu - q_v^{-1}\partial_\mu \varphi_v$, and integrating out $\tilde{a}_\mu$, keeping only zeroth order in derivatives terms of $a_\mu$, we obtain the Lagrangian

$$\mathcal{L} = \frac{1}{2B_0 n_0}\epsilon_{ij}[B^i + q_v^{-1}B_0 \epsilon^{ik}\partial_k \varphi_v]\partial_t[B^j + q_v^{-1}B_0 \epsilon^{jl}\partial_l \varphi_v] + \frac{m_v}{2n_0^2 \bar{\rho}_v}\left[\partial_t(B^i + q_v^{-1}B_0 \epsilon^{ik}\partial_k \varphi_v)\right]^2$$

$$+ \frac{1}{2}\hat{C}_{ij;kl}^{-1}\left[E^{ij} + q_v^{-1}B_0 \delta^{ij}\partial_t \varphi_v\right]\left[E^{kl} + q_v^{-1}B_0 \delta^{kl}\partial_t \varphi_v\right] + A_{ij}J^{ij} + \varphi\rho, \quad (5.3)$$

where the renormalized elastic coefficients take the form

$$\hat{C}_{ij;kl}^{-1} = \frac{1}{8C_1'}P_{ij;kl}^{(0)} + \frac{1}{4C_2}P_{ij;kl}^{(2)}, \qquad (5.4)$$

---

[11]In the triangular lattice the unit cell constitutes two elementary triangular plaquettes, so the flux per unit cell that we consider is an even multiple of the flux quantum.

[12]If the filling fraction $v$ is near but not precisely even-integer-valued, the vortex vacancies effectively experience a small residual background dual magnetic field. In the vortex vacancy condensed phase, we expect this residual dual magnetic field to be repelled from the bulk of a finite system due to the dual Meissner effect resulting in a non-uniform dual magnetic field (boson density) only near the boundary of the sample.

with

$$C_1' = C_1 + \frac{g_v B_0^2}{4q_v^2}. \tag{5.5}$$

We observe that the bulk modulus is increased after vacancies are condensed, while the shear modulus is not renormalized. One can explain this phenomenon by the following argument. The short-range interaction between vacancies in (5.1) generates the mass term for the *scalar potential* $a_t$ in the Higgs phase. Due to the coupling of $a_t$ and the compression part of the elastic sector, the bulk modulus is enhanced after we integrate out the gauge field. In other words, the condensate of vacancies resists the compression of the resulting crystal due to the vacancy-elasticity interaction. An alternative derivation of the renormalization formula of the compression modulus $C_1$ can be found in Appendix C. Curiously, the renormalization found here is opposite to what happens in the finite-temperature vortex supersolid in a three-dimensional type-II superconductor, where is was predicted that a condensation of vacancies/interstitials reduces the bulk modulus [80].

We now redefine the tensor gauge field $\tilde{A}_{ij} = A_{ij} - q_v^{-1} B_0 \delta_{ij} \varphi_v$, and then rewrite the Lagrangian (5.3) as

$$\mathcal{L} = \frac{1}{2B_0 n_0} \epsilon_{ij} \tilde{B}^i \partial_t \tilde{B}^j + \frac{m_v}{2n_0^2 \bar{\rho}_v} \left( \partial_t \tilde{B}^i \right)^2 + \frac{1}{2} \hat{C}_{ij;kl}^{-1} \tilde{E}^{ij} \tilde{E}^{kl} + (\tilde{A}_{ij} + q_v^{-1} B_0 \delta_{ij} \varphi_v) J^{ij} + \varphi \rho , \tag{5.6}$$

with $\tilde{B}^i$ and $\tilde{E}_{ij}$ defined in Eqs. (3.10), (3.11) with $A_{ij}$ being replaced by $\tilde{A}_{ij}$. In the Lagrangian (5.6), $\varphi_v$ is the Lagrange multiplier that imposes the condition $\delta_{ij} J^{ij} = 0$.[13] So in contrast to the original vortex crystal, the vortex supersolid obeys the oridinary glide constraint (3.18). To the lowest order in derivatives the constrained Lagrangian is just

$$\mathcal{L} = \frac{1}{2B_0 n_0} \epsilon_{ij} \tilde{B}^i \partial_t \tilde{B}^j + \frac{1}{2} \hat{C}_{ij;kl}^{-1} \tilde{E}^{ij} \tilde{E}^{kl} + \tilde{A}_{ij} J^{ij} + \varphi \rho , \tag{5.7}$$

which is the dual effective theory of a time-reversal breaking two-dimensional crystal. Elastic dual gauge theory for such a crystal was previously derived in [68,71] by applying a duality transformation to the following elastic Lagrangian

$$\mathcal{L} = \frac{B_0 n_0}{2} \epsilon^{ij} u_i \partial_t u_j - \frac{1}{2} \hat{C}_{ij;kl} u_{ij} u_{kl}. \tag{5.8}$$

In Appendix C we investigate the condensation of vortex defects directly in terms of the effective theory (2.1). In addition, there we introduce an external $U(1)$ gauge potential that couples to the bosonic particle number current. We find that the constituents of the vortex supersolid are neutral under the $U(1)$ bosonic particle number symmetry, but instead carry the magnetic and dipole moments. We also discover that the vortex supersolid does not exhibit the Meissner effect.

Finally, we discuss how our analysis differs from Ref. [71], where condensation of vortices was also investigated in two-dimensional quantum vortex crystals. The key difference that, in contrast to our theory, vortex quasiparticles are assumed to couple to the *full* $u(1)$ dual gauge field in [71]. Since this field has a finite magnetic field background fixed by the superfluid density, condensation of vortices results in the Abrikosov lattice of dual vortices which bind localized elementary bosons to their cores. Based on that, the authors of [71] argued that the resulting chiral crystal is the crystal of localized bosons, which also has the modified glide constraint that follows from the global $U(1)$ particle number symmetry. This would imply that

---

[13]This must be contrasted to a time-reversal invariant quantum supersolid, where the superfluid order completely relaxes the glide constraint [68,70,71].

this crystal cannot undergo quantum melting via the conventional proliferation of dislocations. Our construction that relies on microscopic physics of vacancies and interstitials, however, indicates that the glide constraint of the vortex supersolid crystal is the ordinary one, $\delta_{ij}J^{ij} = 0$, and the conventional defect-assisted continuous quantum melting of such a crystal is possible. Different possibilities of how this can happen will be discussed in detail in Sec. 7.

# 6 Vector gauge theories of vortex crystal and vortex supersolid

In the translation-broken phase, the symmetric tensor gauge theories of the vortex and vortex supersolid crystals derived in Secs. 3 and 5, respectively, are adequate to explain the essential physics. However, in order to understand the dislocation-assisted melting of the vortex supersolid towards a bond-ordered fluid, the bond angle $\theta$ should be promoted to a dynamical field by adding quadratic terms to the Lagrangian (2.1) [68, 69, 73]

$$\mathcal{L} \to \mathcal{L} + \frac{1}{2}(\partial_t\theta)^2 - \frac{K}{2}(\partial_i\theta)^2, \tag{6.1}$$

with $K > 0$. In this section we derive the vector gauge theories of the vortex crystal and the vortex supersolid and discuss their relation to the symmetric tensor gauge theories (3.12), (5.6), respectively.

## 6.1 Vortex crystal

First, we rewrite the Lagrangian (6.1) using the Hubbard-Strantonovich transformation

$$\mathcal{L} = \mathcal{L}_g(a_\mu) + \frac{1}{2B_0n_0}\epsilon^{ij}\tilde{\pi}_i\partial_t\tilde{\pi}_j + \tilde{\pi}_i\partial_t u_i + \frac{1}{2}C^{-1}_{ij;kl}\tilde{\sigma}_{ij}\tilde{\sigma}_{kl} - \tilde{\sigma}_{ij}(\partial_i u_j - \theta\epsilon_{ij}) + B_0 e^i u_i + a_\mu j^\mu_\nu$$

$$+ L\partial_t\theta - \tau_i\partial_i\theta - \frac{1}{2}L^2 + \frac{1}{2K}\tau_i^2. \tag{6.2}$$

Separating the strain field $u_i$ into the singular $u_i^s$ and smooth $u_i^e$ components, and integrating out the smooth part yields the conservation law (3.4). We then perform the field redefinitions (3.6) and introduce the elastic-electromagnetic fields (3.7) that satisfy the Bianchi identity

$$\partial_t B^i + \epsilon_{jk}\partial^j E^{ki} = 0. \tag{6.3}$$

The Bianchi identity (6.3) is automatically satisfied provided one introduces the following vector and tensor gauge fields

$$B^i = \epsilon_{jk}\partial^j A^{ki}, \tag{6.4}$$

$$E_{ij} = -\partial_t A_{ij} + \partial_i A_{0j}. \tag{6.5}$$

Note that the tensor field $A_{ij}$ is *not* symmetric. We now split the bond angle field $\theta = \theta^s + \theta^e$ and integrate out the elastic part $\theta^e$. This gives us the *dynamical Ehrenfest constraint*

$$\partial_t L - \partial_i \tau_i = \epsilon_{ij}\tilde{\sigma}_{ij} = \epsilon_{ij}E_{ij}. \tag{6.6}$$

Using Eqs. (6.4), (6.5), we can solve the dynamical Ehrenfest constrain (6.6) by introducing an additional $u(1)$ gauge field $b_\mu$ [69, 73]

$$L = \epsilon^{ij}\partial_i b_j - \epsilon_{ij}A_{ij}, \tag{6.7}$$

$$\tau_i = \epsilon_{ij}(\partial_t b_j - \partial_j b_t - A_{0j}). \tag{6.8}$$

We then end up with the final dual Lagrangian

$$\mathcal{L} = \mathcal{L}_g(a_\mu) + \frac{1}{2B_0 n_0} \epsilon_{ij}(B^i + B_0 \epsilon^{ik} a_k)\partial_t(B^j + B_0 \epsilon^{jl} a_l) + \frac{1}{2}\tilde{C}^{-1}_{ij;kl}\left(E^{ij} + B_0 \delta^{ij} a_t\right)\left(E^{kl} + B_0 \delta^{kl} a_t\right)$$
$$+ \frac{1}{2K}(\partial_t b_k - \partial_k b_t - A_{0k})^2 - \frac{1}{2}\left(\epsilon^{ij}\partial_i b_j - \epsilon_{ij}A_{ij}\right)^2$$
$$+ A_{ij}J^{ij} + A_{0i}\rho_i + b_i j_d^i + b_t \rho_\theta + a_\mu j_\nu^\mu, \quad (6.9)$$

where we have introduced the dislocation current

$$J^{ij} = \epsilon^{il}\epsilon^{jk}(\partial_l \partial_t - \partial_t \partial_l)u_k^s, \quad (6.10)$$

the (skew) dislocation density,

$$\rho_i = \epsilon^{ij}\epsilon^{lk}\partial_l \partial_k u_j^s = \epsilon^{ij}\chi_j, \quad (6.11)$$

the disclination current

$$j_d^i = \epsilon_{ij}\left(\partial_j \partial_t - \partial_t \partial_j\right)\theta^s, \quad (6.12)$$

the disclination density

$$\rho_\theta = \epsilon^{ij}\partial_i \partial_j \theta^s. \quad (6.13)$$

One may notice the appearance of a new disclination current (6.12). The promotion of the Ehrenfest constraint to the dynamical form (6.6) means that we allow the disclinations to move, which will happen in the bond-ordered fluid phase.

The dual action (6.2) has the following gauge symmetries

$$A_{ij} \to A_{ij} + \partial_i \lambda_j + B_0 \delta_{ij}\beta, \quad A_{0j} \to A_{0j} + \partial_t \lambda_j, \quad (6.14)$$
$$b_i \to b_i + \lambda_i + \partial_i \phi, \quad b_t \to b_t + \partial_t \phi, \quad (6.15)$$
$$a_\mu \to a_\mu + \partial_\mu \beta. \quad (6.16)$$

These gauge symmetries result in the modified glide constraint (3.17), the conservation of the disclination number $\partial_t \rho_d + \partial_i j_d^i = 0$, and

$$\partial_\mu J^{\mu i} - j_d^i = 0. \quad (6.17)$$

The last equation implies that the motion of disclinations (charges) *must* be accompanied by the annihilation or creation of dislocations (dipoles).

In the translation-broken crystal phase, where the disclination current vanishes $j_d^i = 0$, one can perform a $\lambda_i$ gauge transformation to eliminate the field $b_i$ from Eq.(6.9). As a result, the anti-symmetric part of the tensor gauge field, $\epsilon_{ij}A_{ij}$, acquires a mass and hence is eliminated at low energies. After integrating out $A_{0k}$ and renaming $b_t \to \varphi$, we recover the symmetric tensor gauge theory of the vortex lattice (3.12) with the charge density given by

$$\rho = \rho_\theta + \partial_i \rho^i, \quad (6.18)$$

which is consistent with Eq. (3.14). In summary, in the vortex crystal phase at low energies, free disclinations are static, and we can ignore the dynamics of the bond angle field $\theta$. In this phase the two dual theories (3.12) and (6.9) are equivalent at low energies.

## 6.2 Vortex supersolid

Starting from the Langragian of the vector charge theory (6.9), we first replace $a_\mu j_\nu^\mu$ by the vortex's Lagrangian (5.1). We then follow the calculations discussed in Sec. 5 and arrive at the vector charge theory of the vortex supersolid crystal of the form

$$\mathcal{L} = \mathcal{L}_{gauge} + \mathcal{L}_{matter}. \tag{6.19}$$

The Lagrangian of the gauge sector is

$$\mathcal{L}_{gauge} = \frac{1}{2B_0 n_0}\epsilon_{ij}\tilde{B}^i \partial_t \tilde{B}^j + \frac{1}{2}\hat{C}^{-1}_{ij;kl}\tilde{E}^{ij}\tilde{E}^{kl} + \frac{1}{2K}\left(\partial_t b_k - \partial_k b_t - A_{0k}\right)^2 - \frac{1}{2}\left(\epsilon^{ij}\partial_i b_j - \epsilon_{ij}\tilde{A}_{ij}\right)^2, \tag{6.20}$$

where we defined the *elastic-electromagnetic* fields

$$\tilde{B}^i = \epsilon_{jk}\partial^j \tilde{A}^{ki}, \qquad \tilde{E}_{ij} = -\partial_t \tilde{A}_{ij} + \partial_i A_{0j} \tag{6.21}$$

in terms of the gauge-transformed field $\tilde{A}_{ij} = A_{ij} - B_0 \delta_{ij}\varphi_v$. The Lagrangian of the crystalline topological defect sector is given by

$$\mathcal{L}_{matter} = \tilde{A}_{ij}J^{ij} + A_{0i}\rho_i + b_i j_d^i + b_t \rho_d. \tag{6.22}$$

The residual gauge symmetries of (6.19) are

$$\tilde{A}_{ij} \to \tilde{A}_{ij} + \partial_i \lambda_j, \quad A_{0j} \to A_{0j} + \partial_t \lambda_j, \tag{6.23}$$

$$b_i \to b_i + \lambda_i + \partial_i \phi, \quad b_t \to b_t + \partial_t \phi, \tag{6.24}$$

which imply the conservation of the disclination number and the non-conservation of the dislocation number (6.17). In the vortex supersolid phase, the vector gauge theory (6.19) has the ordinary glide constraint (3.18) which is derived exactly as in Sec. 5. In addition in that phase, the vector charge theory (6.19) and scalar charge theory (5.7) are equivalent at low energies.

# 7 Quantum melting of vortex crystal

In this section, we investigate the quantum melting of the vortex crystal using the vector gauge theory (6.9). As argued above, a direct dislocation-assisted continuous conventional melting transition is impossible. Instead, in this paper, we consider a scenario, where the melting of the vortex crystal happens in two steps. First, as already discussed in Sec. (5), the vortex vacancies or interstitials condense giving rise to the vortex supersolid crystal. Second, the latter undergoes a continuous quantum melting transition due to the proliferation of dislocations. In this section, we investigate two types of dislocation-assisted continuous quantum melting of the vortex supersolid towards the nematic and smectic phases.

## 7.1 Quantum melting towards nematic phase

Here we analyze the dislocation-assisted melting mechanism of the vortex supersolid towards the nematic fluid. Although it is natural to expect that the vortex supersolid crystal forms a triangular lattice, here for simplicity, we will discuss the case of the square lattice. We checked that our main predictions do not change qualitatively for the triangular crystal.

First, we introduce bosonic fields that represent dislocations. In particular, $\psi_{\mathbf{d}}$ annihilates a dislocation with the dipole moment $\mathbf{d}$ that correspond to the Burgers vector $\chi_i = \epsilon_{ij}d^j$. In

the following, we will condense the pair $a = 1, 2$ of elementary dislocation dipoles $\mathbf{d}_a$ which are the primitive vectors of the lattice[14]. Following closely [71], we introduce the covariant derivative of $\psi_{\mathbf{d}}$ as[15]

$$D_t \psi_{\mathbf{d}_a} = \left( \partial_t - i d_a^j (A_{0j} - \partial_t b_j) \right) \psi_{\mathbf{d}_a}, \tag{7.1}$$

$$D_i \psi_{\mathbf{d}_a} = \left( \partial_i - i d_a^j (\tilde{A}_{ij} - \partial_i b_j) \right) \psi_{\mathbf{d}_a}, \tag{7.2}$$

with $d_a^j$ being the $j^{\text{th}}$ component of the vector $\mathbf{d}_a$. Notice that as expected the dislocation field couples to the gauge field $b_i$ in the same manner as an electric dipole moment couples to the electromagnetic field. The dislocation field $\psi_{\mathbf{d}_a}$ transforms under the gauge transformations (6.23) and (6.24) as

$$\psi_{\mathbf{d}_a} \to e^{-i d_a^j \partial_j \phi} \psi_{\mathbf{d}_a}. \tag{7.3}$$

In order to analyze the Higgs mechanism of the quantum melting, we replace the matter Lagrangian in (6.19) with the dynamical Lagrangian for dislocations [71]

$$\mathcal{L}_{dis} = \sum_{a=1,2} \left( i \psi_{\mathbf{d}_a}^\dagger D_t \psi_{\mathbf{d}_a} - \frac{1}{2m_d} |\Pi_{\mathbf{d}_a}^{jk} D_k \psi_{\mathbf{d}_a}|^2 \right) - V_{dis}(\psi_{\mathbf{d}_a}), \tag{7.4}$$

where $\Pi_{\mathbf{d}_a}^{jk} = \delta^{jk} - \frac{d_a^j d_a^k}{|d_a|^2}$ is the projection to direction perpendicular to the dipole vector $\mathbf{d}_a$. Due to the projection, the ordinary glide constraint $\delta_{ij} J^{ij} = 0$ is satisfied automatically. The potential term $V_{dis}$ must be invariant under the gauge transformation (7.3) and point group symmetry transformations of the lattice. For simplicity, in this subsection it is chosen to be[16]

$$V_{dis} = \sum_{a=1,2} \left( \mu \psi_{\mathbf{d}_a}^\dagger \psi_{\mathbf{d}_a} + \frac{\mathfrak{g}}{2} |\psi_{\mathbf{d}_a}|^4 \right). \tag{7.5}$$

The short-range part of interaction between dislocations is assumed to be repulsive, $\mathfrak{g} > 0$. If one now tunes the chemical potential $\mu$ to negative values, the dislocations acquire a finite expectation value $\bar{\rho}_{\mathbf{d}_1} = \bar{\rho}_{\mathbf{d}_2} = \bar{\rho} = \frac{|\mu|}{\mathfrak{g}}$. As a consequence, both $\psi_{\mathbf{d}_1}$ and $\psi_{\mathbf{d}_2}$ condense and the vortex supersolid crystal melts into the quantum nematic fluid.

Now we rewrite the dislocation fields as $\psi_{\mathbf{d}_a} = \sqrt{\bar{\rho} + \sigma_a} e^{-i\varphi_a}$ with the gauge transformation (7.3) acting on the phase $\varphi_a$ as

$$\varphi_a \to \varphi_a + d_a^j \partial_j \phi. \tag{7.6}$$

After integrating out the heavy fields $\sigma_i$, using Appendix B, we arrive at the quadratic Lagrangian

$$\mathcal{L}_{dis} \to \mathcal{L}_{\varphi_a} = \bar{\rho} d (A_{01} + A_{02}) + \frac{d^2}{2\mathfrak{g}} \left( (\partial_t \tilde{\varphi}_1 + (A_{01} - \partial_t b_1))^2 + (\partial_t \tilde{\varphi}_2 + (A_{02} - \partial_t b_2))^2 \right)$$

$$- \frac{\bar{\rho} d^2}{2m_d} \left( (\partial_2 \tilde{\varphi}_1 + (\tilde{A}_{21} - \partial_2 b_1))^2 + (\partial_1 \tilde{\varphi}_2 + (\tilde{A}_{12} - \partial_1 b_2))^2 \right), \tag{7.7}$$

---

[14]For a square lattice the simplest choice is to point the cartesian coordinates along the primitive vectors and hence $\mathbf{d}_1 = d\hat{x}$ and $\mathbf{d}_2 = d\hat{y}$, where $d$ is the lattice spacing.

[15]In Ref. [71], Kumar and Potter introduced the coupling of the dislocation field to the symmetric tensor gauge fields in the scalar charge theory. Here instead, we couple dislocations to gauge fields in the vector charge theory (6.20). One can check that in the lattice phase, where $b_i \to 0$, one recovers the covariant derivative of [71].

[16]One can consider a more general form of the potential, however, as long as it is minimized by $\bar{\rho}_{\mathbf{d}_1} = \bar{\rho}_{\mathbf{d}_2} \neq 0$, the Higgs mechanism analysis will be qualitatively the same as discussed in this subsection.

where we introduced $\tilde{\varphi}_i = \varphi_i/d$. We observe that, $A_{0i}$, $\tilde{A}_{12}$ and $\tilde{A}_{21}$ are gapped out due to the Higgs mechanism. Using $\lambda_i$, we fix now the gauge to $\tilde{A}_{11} = \tilde{A}_{22} = 0$ and integrate out the massive fields to obtain the effective Lagrangian of the quantum nematic phase [17]

$$\mathcal{L} = \alpha_1 \tilde{e}_i^2 - \alpha_2 \tilde{b}^2, \tag{7.8}$$

with $\tilde{e}_i = \partial_i b_t - \partial_t \tilde{\varphi}_i$ and $\tilde{b} = \epsilon^{ij} \partial_i \tilde{\varphi}_j$. The coefficient $\alpha_1$ and $\alpha_2$ in Eq. (7.8) are found to be

$$\alpha_1 = \frac{(\mathfrak{g}/d^2 + K)}{2(\mathfrak{g}/d^2 - K)^2}, \qquad \alpha_2 = \frac{\bar{\rho}}{2(\bar{\rho} + 2m_d/d^2)}. \tag{7.9}$$

We observe that the fields $\tilde{\varphi}_i$ and $b_t$ are gauge potentials of a new emergent $u(1)$ gauge theory with the gauge redundancy

$$b_t \to b_t + \partial_t \phi, \qquad \tilde{\varphi}_i \to \tilde{\varphi}_i + \partial_i \phi, \tag{7.10}$$

that one can obtain from Eqs. (6.24) and (7.6).

   We ended up with the Lagrangian (7.8) which encodes the dual gauge description of an ordinary time-reversal invariant two-dimensional quantum nematic phase that breaks spontaneously $SO(2)$ rotation symmetry. Note that although we have started from the chiral theory of the vortex supersolid, the resulting nematic model appears from Eq. (7.8) to be *non-chiral*. Although time-reversal and parity breaking terms are generated, they are higher-order in the derivative expansion and thus were not included in the leading order Lagrangian (7.8). Thus at small wave vectors and frequencies, the vortex chiral nematic phase cannot be distinguished from the ordinary nematic phase.

## 7.2 Quantum melting towards smectic phase

In this subsection, we will show how to obtain the effective theory of the vortex chiral smectic phase by proliferating only one component of the dislocation field. Our computation is similar in spirit to the one done in the previous subsection.

   In this case, we start from the dislocation Lagrangian

$$\mathcal{L}_{dis} = \sum_{a=1,2} \left( i\psi_{\mathbf{d}_a}^\dagger D_t \psi_{\mathbf{d}_a} - \frac{1}{2m_d} |\Pi_{\mathbf{d}_a}^{jk} D_k \psi_{\mathbf{d}_a}|^2 \right) - V'_{dis}(\psi_{\mathbf{d}_i}), \tag{7.11}$$

with the dislocation potential [68][18]

$$V'_{dis} = \sum_{a=1,2} \left( \mu \psi_{\mathbf{d}_a}^\dagger \psi_{\mathbf{d}_a} + \frac{\mathfrak{g}}{2} |\psi_{\mathbf{d}_a}|^4 \right) + \mathfrak{g}' |\psi_{\mathbf{d}_1}|^2 |\psi_{\mathbf{d}_2}|^2, \tag{7.12}$$

with $\mathfrak{g}' > \mathfrak{g} > 0$. We again will set $\mathbf{d}_1 = d\hat{x}, \mathbf{d}_2 = d\hat{y}$. When the chemical potential $\mu$ becomes negative, the dislocations condense. However, for $\mathfrak{g}' > \mathfrak{g}$ the potential $V'_{dis}$ is minimized only when one component of the dislocation field acquires a finite expectation value. After spontaneously choosing $\langle \psi_{\mathbf{d}_1} \rangle \neq 0$, we rewrite the dislocation field as $\psi_{\mathbf{d}_1} = \sqrt{\bar{\rho} + \sigma_1} e^{-i\varphi_1}$ with $\bar{\rho} = \frac{|\mu|}{\mathfrak{g}}$. After integrating out $\sigma_1$ one obtains the Lagrangian

$$\mathcal{L}_{dis} = \mathcal{L}_{\varphi_1} + \mathcal{L}'_{\psi_{\mathbf{d}_2}}, \tag{7.13}$$

---

[17]We ignored the higher derivative of $A_{0i}$, $\tilde{A}_{12}$ and $\tilde{A}_{21}$ in Eq. (6.19).

[18]The potential (7.5) is the $\mathfrak{g}' \to 0$ limit of the potential (7.12). As commented in the Ref. [68], the nematic and smectic phase appears when $\mathfrak{g} > \mathfrak{g}' \geq 0$ and $\mathfrak{g}' > \mathfrak{g} > 0$, respectively. In the previous subsection we chose $\mathfrak{g}' = 0$ to simplify the calculations. One can repeat the calculation for the nematic phase with a general value $\mathfrak{g} > \mathfrak{g}' \geq 0$ and obtain the same final result.

where the Lagrangian of the Goldstone mode $\varphi_1$ is

$$\mathcal{L}_{\varphi_1} = \bar{\rho} d A_{01} + \frac{d^2}{2\mathfrak{g}} (\partial_t \tilde{\varphi}_1 + (A_{01} - \partial_t b_1))^2 - \frac{\bar{\rho} d^2}{2m_d} (\partial_2 \tilde{\varphi}_1 + (\tilde{A}_{21} - \partial_2 b_1))^2, \tag{7.14}$$

with $\tilde{\varphi}_1 = \varphi_1/d$. On the other hand, the dislocation field $\psi_{\mathbf{d}_2}$ acquires a positive effective chemical potential $\mu' = \mu(1 - \frac{\mathfrak{g}'}{\mathfrak{g}})$ and thus

$$\mathcal{L}'_{\psi_{\mathbf{d}_2}} = i\psi^{\dagger}_{\mathbf{d}_2} D_t \psi_{\mathbf{d}_2} - \frac{1}{2m_d} |\Pi^{jk}_{\mathbf{d}_2} D_k \psi_{\mathbf{d}_2}|^2 - \mu' |\psi_{\mathbf{d}_2}|^2. \tag{7.15}$$

As a result, the dislocation field $\psi_{\mathbf{d}_2}$ is gapped and the above Lagrangian can be ignored at low energies.

We observe that both $A_{01}$ and $\tilde{A}_{21}$ acquire a mass due to the Higgs mechanism. We then use the gauge transformation $\lambda_1$ to eliminate $\tilde{A}_{11}$, integrate out the massive fields $A_{01}$, $\tilde{A}_{21}$ and keep the lowest gradient terms to obtain the effective action of the vortex chiral smectic phase[19]

$$\mathcal{L} = \frac{1}{8C_2} \left( \partial_t \vec{A} - \vec{\nabla} A_0 \right)^2 + \frac{1}{16C'_1} (\partial_t A_2 - \partial_2 A_0)^2 + \frac{1}{2K} \left[ (\partial_t \tilde{\varphi}_1 - \partial_1 b_t)^2 + (\partial_t b_2 - \partial_2 b_t - A_0)^2 \right]$$
$$- \frac{1}{2} (\partial_1 b_2 - \partial_2 \tilde{\varphi}_1 - A_1)^2, \tag{7.16}$$

where to simplify notation we renamed

$$\tilde{A}_{i2} \rightarrow A_i, \qquad A_{02} \rightarrow A_0. \tag{7.17}$$

The residual gauge symmetries are

$$A_\mu \rightarrow A_\mu + \partial_\mu \lambda_2, \tag{7.18}$$

$$\tilde{\varphi}_1 \rightarrow \tilde{\varphi}_1 + \partial_1 \phi, \qquad b_2 \rightarrow b_2 + \lambda_2 + \partial_2 \phi, \qquad b_t \rightarrow b_t + \partial_t \phi. \tag{7.19}$$

# 8  Discussion and outlook

With the help of the fracton-elasticity duality, a comprehensive understanding of different time-reversal invariant quantum phases (and allowed quantum phase transitions between these phases) of two-dimensional bosonic matter has been achieved recently [68, 70, 71]. Inspired by these ideas, in this paper we made a step towards a better understanding of quantum vortex crystals and some related time-reversal breaking quantum phases of zero-temperature two-dimensional bosons in an external magnetic field.

The physical properties of the two-dimensional quantum vortex supersolid phase, where translation symmetry is broken spontaneously and vortex vacancy/interstitial defects are condensed, are currently not well understood. In particular, it is unclear to which extent it is different from a magneto-crystal, i.e., a crystal of bosonic atoms in an external magnetic field. In future work, it would be interesting to compute the particle number, momentum and energy transport properties of this phase and compare them with the linear response of the magneto-crystal. These results will be also worth comparing with predictions of transport of

---

[19]To simplify the coefficients of different terms of the Lagrangian (7.16) here we assumed $K \gg \frac{\mathfrak{g}}{d^2}$ and $\frac{\bar{\rho} d^2}{m_d} \gg 1$. This limit can be achieved by taking $\mathfrak{g} \rightarrow 0$. Gaussian integration now simply corresponds to $A_{01} \rightarrow \partial_t b_1 - \partial_t \tilde{\varphi}_1$, $\tilde{A}_{21} \rightarrow \partial_2 b_1 - \partial_2 \tilde{\varphi}_1$.

finite-temperature vortex supersolids in type-II superconductors in external magnetic fields investigated in Refs. [79, 80].

It would be important to find out if the vortex supersolid phase discussed in this paper occurs for superfluid bosons with short-range or dipolar interactions in a constant artificial magnetic field (induced by rotation or some other mechanism [15–17, 83]). This may be realized and investigated in future cold atom experiments. A finite-difference of the bulk moduli of the vortex crystal and vortex supersolid predicted in this paper can be in principle utilized to distinguish these two crystalline phases of matter.

In a recent preprint [89], some evidence of hexatic and isotropic quantum vortex fluids was reported in extremely weakly pinned a-MoGe superconductor films at very low temperatures, see however [90]. We believe that it is worth searching experimentally for the quantum vortex supersolid phase in that system.

Finally, one can apply the formalism developed in this paper to other crystals with long-range interactions, where the elastic sector couples to Hubbard-Stratonovich fields that mediate long-range forces[20]. As in this work, it is straightforward to extract low energy modes and static interaction potentials between topological defects in the crystalline phase. We expect that the analysis of the quantum melting transitions might be sensitive to microscopic details of the studied crystal.

**Note added.**— Recently we became aware of a related work on a duality between fractons and two-dimensional quantum smectics [91–93].

## Acknowledgements

We acknowledge discussions with Grigory Volovik. We thank Ajesh Kumar and Andrew Potter for comments on the manuscipt. DXN was supported by Brown Theoretical Physics Center. AG was supported by the Brown University. The work of SM is funded by the Deutsche Forschungsgemeinschaft (DFG, German Research Foundation) under Emmy Noether Programme grant no. MO 3013/1-1 and under Germany's Excellence Strategy - EXC-2111 - 390814868.

## A   Stability analysis of vortex crystal

Here we perform a stability analysis of the ground state of a vortex crystal. Our starting point is the quadratic effective theory of Watanabe and Murayama [31]

$$\mathcal{L} = \frac{n_0}{2mc_s^2}\Big[\dot{\varphi}^2 - c_s^2\big(\partial_i\varphi + B_0\epsilon_{ij}u^j\big)^2\Big] - \frac{B_0 n_0}{2}\epsilon_{ij}u_i\dot{u}_j - \mathcal{E}_{\mathrm{el}}(u^{ij}), \tag{A.1}$$

whose degrees of freedom are the smooth part of the superfluid phase $\varphi$ and the displacements $u^i$. This theory is completely equivalent to the effective theory (2.1), see [32], and is a more convenient formulation for the purpose of the stability analysis.

The corresponding Hamiltonian density is

$$\mathcal{H} = \pi_\varphi\dot{\varphi} + \pi_{u^i}\dot{u}^i - \mathcal{L}_{\mathrm{eff}} = \frac{n_0}{2mc_s^2}\dot{\varphi}^2 + \frac{n_0}{2m}\big(\partial_i\varphi + B_0\epsilon_{ij}u^j\big)^2 + \mathcal{E}_{\mathrm{el}}(u^{ij}). \tag{A.2}$$

Since we are interested in the stability of the vortex crystal ground state, we restrict our attention to static configurations whose effective Hamiltonian is

$$H = \int d^2x\left[\frac{n_0}{2m}\big(\partial_i\varphi + B_0\epsilon_{ij}u^j\big)^2 + \mathcal{E}_{\mathrm{el}}(u^{ij})\right]. \tag{A.3}$$

---

[20]In our model of the superfluid vortex crystal, the boson-vortex duality ensures that the proper Hubbard-Stratonovich fields are $u(1)$ gauge fields $a_\mu$

The ground state is stable if the matrix of second variational derivatives with respect to all fields[21] $\Phi = (B_0^{-1/2}\varphi, u^x, u^y)$ is positive definite. In momentum space

$$M^{kl}(\mathbf{q}) = \frac{1}{2}\frac{\delta^2 H}{\delta\Phi^k(-q)\Phi^l(q)}, \tag{A.4}$$

with $k, l = 1, 2, 3$. Low-momentum expansion of the eigenvalues of this matrix, which all must be positive, gives rise to the condition $C_2 > 0$, but does not put any constraints on the sign of the compression modulus $C_1$. We conclude that in contrast to a time-reversal invariant crystal, the compression modulus does not have to be positive in a vortex crystal. This is in agreement with previous studies [18–20, 26, 81].

# B  Higgs mechanism in non-relativistic field theory

We consider the Lagrangian of a non-relativistic field theory

$$\mathcal{L} = i\psi^\dagger D_t\psi - \frac{1}{2m}|D_i\psi|^2 - \mu\psi^\dagger\psi - \frac{\mathfrak{g}}{2}(\psi^\dagger\psi)^2, \tag{B.1}$$

where we defined the covariant derivative as

$$D_t = \partial_t - ia_t, \qquad D_i = \partial_i - ia_i. \tag{B.2}$$

To have a stable ground state, we restrict our attention to $\mathfrak{g} > 0$ in the potential term of (B.1). When the chemical potential $\mu$ changes its sign from positive to negative, the bosons are condensed and develop a finite vacuum expectation value

$$\langle\psi^\dagger\psi\rangle = \bar{\rho} = \frac{|\mu|}{\mathfrak{g}}. \tag{B.3}$$

We then express the boson field as

$$\psi = \sqrt{\bar{\rho}+\sigma}\,e^{i\varphi}. \tag{B.4}$$

Ignoring the total derivative terms as well as higher order derivative terms of $\sigma$, one can rewrite the effective Lagrangian in terms of $\sigma$ and $\varphi$

$$\mathcal{L} = a_t\bar{\rho} - \sigma(\partial_t\varphi - a_t) - \frac{\bar{\rho}+\sigma}{2m}(\partial_i\varphi - a_i)^2 - \frac{\mathfrak{g}}{2}\sigma^2, \tag{B.5}$$

we then integrate out $\sigma$ and keep only the lowest order terms up to second order in $\varphi$ to yield

$$\mathcal{L} = a_t\bar{\rho} + \frac{1}{2\mathfrak{g}}(\partial_t\varphi - a_t)^2 - \frac{\bar{\rho}}{2m}(\partial_i\varphi - a_i)^2. \tag{B.6}$$

The effective action of the Goldstone mode in the non-relativistic theory differs from the effective action in a relativistic superfluid. In the relativistic theory, the coefficients of the time derivative and space derivative terms are the same due to Lorentz invariance. In this paper, we will use this analysis to study the Higgs mechanisms due to condensation of vortex and dislocation matter in the vortex lattice.

---

[21]The factor $B_0^{-1/2}$ in the first component of $\Phi$ ensures that all components of $\Phi$ have the same physical dimension.

## C Vortex defect condensation in effective theory (2.1)

Here, similar to Sec. 2, we start from the effective theory of the vortex crystal developed in [32, 33], whose Lagrangian is

$$\mathcal{L} = \mathcal{L}_g(a_\mu) + \mathcal{L}_{el}(u_i) + B_0 e_i u^i + \mathcal{L}_v(\psi_v) + \epsilon^{\mu\nu\rho} \mathcal{A}_\mu \partial_\nu a_\rho, \tag{C.1}$$

where $a_\mu$ is the dual $u(1)$ gauge field encoding fluctuations around a static background $\bar{a}_\mu$ with the magnetic field $\bar{b} = \epsilon_{ij}\partial_i\bar{a}_j$ fixed by the bosonic density $n_0$. $\mathcal{L}_g(a_\mu)$ is some quadratic Lagrangian of the gauge field, whose precise form will not be important in calculations done in this Appendix. The chiral elasticity Lagrangian is

$$\mathcal{L}_{el} = -\frac{B_0 n_0}{2}\epsilon_{ij} u^i \partial_t u^j - \frac{1}{2}C_{ij;kl}u_{ij}u_{kl}. \tag{C.2}$$

Following the arguments presented in Sec. 5, the quasiparticle vortex Lagrangian is assumed to have the form

$$\mathcal{L}_v = i\psi_v^\dagger D_t \psi_v - \frac{1}{2m_v}|D_i\psi_v|^2 - \mu_v\psi_v^\dagger\psi_v - \frac{g_v}{2}(\psi_v^\dagger\psi_v)^2, \qquad D_\mu = \partial_\mu - iq_v a_\mu, \tag{C.3}$$

where $q_v = \pm 2\pi$ is the charge of the elementary quasiparticle vortex interstitial and vacancy, respectively, with respect to the dual gauge field $a_\mu$. We emphasize that the vortex excitation couples only to $a_\mu$ and not to the background $\bar{a}_\mu$. Finally, in this Appendix we couple the effective theory to the external particle number $U(1)$ background source $\mathcal{A}_\mu$ which is set to vanishes in the ground vortex crystal state [32, 33]. In terms of the full source $A_\mu$ and the background source $\bar{A}_\mu$ (which induces the constant magnetic field $B_0$), $\mathcal{A}_\mu = A_\mu - \bar{A}_\mu$.[22]

We are interested here in how the vortex defect condensation discussed in Sec. 5 can be described directly within the effective theory (C.1). To induce vortex quasiparticle condensation, we change the vortex chemical potential $\mu_v$ from positive to negative values. Following the arguments of Sec. 5, in the vortex condensed phase the vortex matter Lagrangian reduces to the quadratic form

$$\mathcal{L}_v = q_v a_t \bar{\rho}_v + \frac{1}{2g_v}(\partial_t\varphi_v - q_v a_t)^2 - \frac{\bar{\rho}_v}{2m_v}(\partial_i\varphi_v - q_v a_i)^2, \tag{C.4}$$

where $\varphi_v$ is the phase of the vortex field $\psi_v$ and we dropped all total derivative terms such as the vortex Berry term $\sim \bar{\rho}_v\partial_t\varphi_v$.

Now we do a regular gauge transformation and introduce $\tilde{a}_\mu = a_\mu - q_v^{-1}\partial_\mu\varphi_v$. In terms of $\tilde{a}_\mu$ the Lagrangian (C.1) becomes

$$\mathcal{L} = \mathcal{L}_g(\tilde{a}_\mu) + \mathcal{L}_{el}(u_i) + B_0 u^i (\partial_t\tilde{a}_i - \partial_i\tilde{a}_t)$$
$$+ q_v\tilde{a}_t\bar{\rho}_v + \frac{q_v^2}{2g_v}\tilde{a}_t^2 - \frac{q_v^2\bar{\rho}_v}{2m_v}\tilde{a}_i^2 + \epsilon^{\mu\nu\rho}\mathcal{A}_\mu\partial_\nu\tilde{a}_\rho. \tag{C.5}$$

Since $\tilde{a}_\mu$ is smooth, we will now integrate it out. We will perform the Gaussian integration over $\tilde{a}_\mu$ keeping only terms that are up to and including first order in derivatives.[23] The result is

$$\mathcal{L} = -\frac{B_0 n_0}{2}\epsilon_{ij} u^i \partial_t u^j - \frac{1}{2}C_{ij;kl}u_{ij}u_{kl} - \frac{1}{2}\frac{g_v(\mathcal{B} + q_v\bar{\rho}_v + B_0\partial_i u^i)^2}{q_v^2} + \frac{m_v}{2q_v^2\bar{\rho}_v}(\epsilon_{ij}\mathcal{E}_j + B_0\partial_t u_i)^2. \tag{C.6}$$

---

[22]The source $\mathcal{A}_\mu$ couples also to the background $\bar{a}_\mu$ via $\epsilon^{\mu\nu\rho}\mathcal{A}_\mu\partial_\nu\bar{a}_\rho$. This background term is not explicitly included in (C.1) since its presence plays no role in the vortex condensation calculation.

[23]Every term that contributes to $\mathcal{L}_g(\tilde{a}_\mu)$ contains at least two derivatives [32, 33] and thus we can completely neglect the gauge sector Lagrangian $\mathcal{L}_g$ in this calculation.

Notice that in contrast to the coupling $\mathcal{E}_i u^i$ expected in an ordinary $U(1)$ charged crystal, here we have the coupling terms $\sim \mathcal{B}\partial_i u^i$ and $\epsilon_{ij}\partial_t u_i \mathcal{E}_j$. As a result, the constituents of the crystal carry no $U(1)$ charge. Instead, away from the crystalline ground state finite fluctuations in the $U(1)$ magnetic moment density $\sim \partial_i u^i$ and the $U(1)$ dipole moment density $\sim \epsilon_{ij}\dot{u}_j$ are induced. The former tells us that the constituents carry circulating bosonic currents around them.

In the following the last term will be dropped because it contains $(\partial_t u_i)^2$ which is subleading[24] compared to the first term in the Lagrangian (C.6), so we will study the effective theory encoded in the Lagrangian

$$\mathcal{L} = -\frac{B_0 n_0}{2}\epsilon_{ij}u^i\partial_t u^j - \frac{1}{2}C_{ij;kl}u_{ij}u_{kl} - \frac{1}{2}\frac{g_v(\mathcal{B}+q_v\bar{\rho}_v+B_0\partial_i u^i)^2}{q_v^2}. \tag{C.7}$$

First, we notice that the last term contains $(\partial_i u^i)^2$ and thus the compression modulus of the vortex supersolid crystal is larger than $C_1$ of the vortex crystal

$$C_1 \to C_1 + \frac{g_v}{4q_v^2}B_0^2, \tag{C.8}$$

which exactly reproduces the result derived in Sec. 5. Second, using now the tensor structure of the elasticity tensor

$$C_{ij;kl} = 8C_1 P^{(0)}_{ij;kl} + 4C_2 P^{(2)}_{ij;kl}, \tag{C.9}$$

with

$$P^{(0)}_{ij;kl} = \frac{1}{2}\delta_{ij}\delta_{kl}, \qquad P^{(2)}_{ij;kl} = \frac{1}{2}\left(\delta_{ik}\delta_{jl} + \delta_{il}\delta_{jk} - \delta_{ij}\delta_{kl}\right), \tag{C.10}$$

we integrate out the displacements $u^i$ and calculate the induced action $S[\mathcal{A}_\mu]$ that is encoded in the following Lagrangian written in Fourier space

$$\mathcal{L}_{ind} = -\frac{g_v}{2q_v^2}(\mathcal{B}_{-q}+q_v\bar{\rho}_v)(\mathcal{B}_q+q_v\bar{\rho}_v) - \frac{1}{2}\frac{g_v^2 B_0^2}{q_v^4}\mathcal{B}_{-q}q_i G_q^{ij}q_j\mathcal{B}_q, \tag{C.11}$$

where $q = (\omega, \mathbf{q})$, $\mathcal{B}$ is the magnetic field constructed from $\mathcal{A}_i$ and the inverse of the elastic propagator $G_q^{ij}$ can be extracted directly from the Lagrangian (C.7)

$$G_q^{-1} = \begin{pmatrix} 4C_1 q_x^2 + 2C_2\mathbf{q}^2 & 4C_1 q_x q_y + iB_0 n_0\omega \\ 4C_1 q_x q_y - iB_0 n_0\omega & 4C_1 q_y^2 + 2C_2\mathbf{q}^2 \end{pmatrix}. \tag{C.12}$$

Here for the compression modulus $C_1$ we must substitute the renormalized value (C.8). Inverting the matrix (C.12) and substituting into Eq. (C.11), we find

$$\mathcal{L}_{ind} = -\frac{g_v}{2q_v^2}(\mathcal{B}_{-q}+q_v\bar{\rho}_v)(\mathcal{B}_q+q_v\bar{\rho}_v) + \frac{1}{2}\frac{g_v^2 B_0^2}{q_v^4}\mathcal{B}_{-q}\frac{2C_2\mathbf{q}^4}{B_0^2 n_0^2\omega^2 - 4C_2(2C_1+C_2)\mathbf{q}^4}\mathcal{B}_q. \tag{C.13}$$

From the denominator of that term we can extract the quadratic dispersion relation of the collective magnetophonon mode supported by the vortex supersolid crystal.

Introducing the London response function as the term $-\mathcal{B}_{-q}\rho_L(\omega,\mathbf{q})\mathcal{B}_q/2$ in the induced Lagrangian, we find

$$\rho_L(\omega,\mathbf{q}) = \frac{g_v}{q_v^2} - \frac{g_v^2 B_0^2}{q_v^4}\frac{2C_2\mathbf{q}^4}{B_0^2 n_0^2\omega^2 - 4C_2(2C_1+C_2)\mathbf{q}^4}. \tag{C.14}$$

Contrary to a superfluid which in the static regime $\omega = 0$ has $\rho_L \sim 1/\mathbf{q}^2$ (that implies the Meissner effect in a superconductor), here $\rho_L \to$ const as in a conventional $U(1)$ insulator. The electric term, which was dropped above will contribute to the conductivity tensor (not computed here), but will not affect the London response.

---

[24]Due to the quadratic dispersion relation of the magnetophonon, we count $\partial_i \sim O(\epsilon)$, but $\partial_t \sim O(\epsilon^2)$.

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
