# Peer review of "Fracton-elasticity duality of two-dimensional superfluid vortex crystals: defect interactions and quantum melting"

_SciPost Physics, doi:SciPost Phys. 9, 076 (2020)_

## Round 2 · Referee Report · Anonymous (Referee 1) · 2020-9-20

Report

This paper contains an impressive discussion and derivation of a dual theory of melting of a vortex crystal in a two-dimensional quantum boson fluid. Perhaps the only missing ingredient is a renormalization group (RG) analysis, similarly to the one employed in the classical field theory of crystal melting, as discussed long time ago by B. Halperin and D. Nelson and by Kleinert in Ref. 31. A generalization of these ideas for a non-relativistic quantum system would be desirable, but not required for this manuscript.

I did not find any obvious flaw in this manuscript and so I don't judge any changes to be necessary. Perhaps the authors can also include in the literature some other earlier works on the classical, non-fractonic counterpart, like Nelson, Young, and others, who also show the subtleties involved when an RG analysis is performed, which is akin to the one in BKT systems. Kleinert's book cited in Ref. 31 included a thorough reference account and references to these earlier works.
  • validity: -
  • significance: -
  • originality: -
  • clarity: -
  • formatting: -
  • grammar: -

Author:  Sergej Moroz  on 2020-10-13  [id 1001]

(in reply to Report 1 on 2020-09-20)

We thank the referee for a careful reading of our manuscript and for constructive criticism, which have helped us to improve our work. The referee gave us several detailed comments, which we respond to in detail below:

-“Perhaps the authors can also include in the literature some other earlier works on the classical, non-fractonic counterpart, like Nelson, Young, and others, who also show the subtleties involved when an RG analysis is performed, which is akin to the one in BKT systems.”

We added a paragraph to the introduction section, where we briefly summarize the historical development of the RG method for the thermal melting transition of two-dimensional crystals. We also refer to the original works by Kosterlitz and Thouless, and the follow-up works by Halperin, Nelson and Young accordingly. 

-“Often a dilute superfluid vortex lattice is treated as a classical Coulomb gas. The paper, would benefit if there is a broader discussion of what aspects of the discussion hold and what do not hold for other non-vortex-based crystals with long-range interacting objects.”

We thank the referee for the suggestion. We added a paragraph and the footnote[20] in the end of the section 8. We discuss the extension of the formalism of fracton/elasticity duality to other crystals with long-range interactions.

---

## Round 2 · Referee Report · Anonymous (Referee 2) · 2020-9-27

Report

The MS uses the techniques that recently became popular to study the classical problem of melting and structure of vortex lattice in a superfluid system at zero temperature. The MS describes the interaction between defects in such a lattice and reports to be short-range, both attractive and repulsive. The MS also discusses the condensation of the defects and supersolidity.

The MS refers to duality discussion in the context of supersolid Helium Ref 32. I think the way it is formulated is misleading because standard supersolidity in Helium was disproven and corresponding claims in experimental articles retracted by Chan et al in follow-up papers. One may still call supersolid the superfluid dislocation but not in the same sense. Here the presentation should be made more accurate.

The authors also refer to superfluid in the magnetic field. This is not a real magnetic field that the authors consider. The corresponding formulations could be better.

The term magneto-crystal appears without definition.

In my opinion, the MS makes, in my opinion, an unjustified bold claim about realization:
"This setup can be realized and investigated in nowadays cold atom experiments." These systems have trap potential. Although box traps are possible, they are finite, and it is difficult to achieve a rapid rotation. Could the authors justify that better or soften the formulation?

Apart from these minor comments, I think it is very good work that should be published.
  • validity: -
  • significance: -
  • originality: -
  • clarity: -
  • formatting: -
  • grammar: -

Author:  Sergej Moroz  on 2020-10-13  [id 1000]

(in reply to Report 2 on 2020-09-27)

We thank the referee for a careful reading of our manuscript and for constructive criticism, which have helped us to improve our work. The referee gave us several comments, which we respond to in detail below:

-“The MS refers to duality discussion in the context of supersolid Helium Ref 32. I think the way it is formulated is misleading because standard supersolidity in Helium was disproven and corresponding claims in experimental articles retracted by Chan et al in follow-up papers. One may still call supersolid the superfluid dislocation but not in the same sense. Here the presentation should be made more accurate.”

We thank the referee for raising an important point. We removed “in the context of supersolid helium” in this sentence. 

-“The authors also refer to superfluid in the magnetic field. This is not a real magnetic field that the authors consider. The corresponding formulations could be better.”

It is correct, the “magnetic field” that we introduced in the manuscript is not a real magnetic field. We added the footnote [4] and several references therein to clarify this point. 

“The term magneto-crystal appears without definition.” 

We added a definition of the magneto-crystal in the second paragraph of section 8. 

“In my opinion, the MS makes, in my opinion, an unjustified bold claim about realization:
 "This setup can be realized and investigated in nowadays cold atom experiments." These systems have trap potential. Although box traps are possible, they are finite, and it is difficult to achieve a rapid rotation. Could the authors justify that better or soften the formulation?”

We softened the statement following the suggestion of the referee.

“Apart from these minor comments, I think it is very good work that should be published.”

We thank the referee again for reviewing our work and providing valuable suggestions.

---

## Round 3 · Author Response

Dear Editor,
Thank you for considering our paper for publication in SciPost Physics.
We thank the referees for careful readings of our manuscript and for their constructive comments, which have helped us to improve our work.
with best regards,
Dung Xuan Nguyen, Andrey Gromov, Sergej Moroz
Thank you for considering our paper for publication in SciPost Physics.
We thank the referees for careful readings of our manuscript and for their constructive comments, which have helped us to improve our work.
with best regards,
Dung Xuan Nguyen, Andrey Gromov, Sergej Moroz

---

## Editorial Decision

published